# Reduplication in Kua'nsi

## Huade Huang

School of Culture, History and Language, The Australian National University, Canberra, ACT 2601, Australia; huade.huang@anu.edu.au

**Abstract:** This paper investigates reduplication in Kua'nsi, a Central Ngwi language of the Sino-Tibetan family, spoken in Yunnan Province, China, by around 5000 speakers. Reduplication is a productive morphological device in Kua'nsi and has complex forms and functions. Although Kua'nsi reduplication shows some similarities with reduplication in other Ngwi languages, it also has reduplicative forms and functions that appear to be cross-linguistically rare. Formally, reduplication in Kua'nsi can be full, partial, or discontinuous. Functionally, it can be used with inflectional and derivational meanings as well as without any semantic or syntactic effect in certain constructions. Some functions of Kua'nsi reduplication appear to be not frequently found across languages. The forms and functions of Kua'nsi reduplication are complex and there is not a one-to-one relationship between the form and function of particular patterns of reduplication.

**Keywords:** Kua'nsi; Ngwi language; reduplication; morphology; Sino-Tibetan

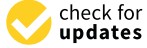



## 1. Introduction

Kua'nsi is a Central Ngwi (also known as Loloish or Yi) language of the Sino-Tibetan family (Bradley 1997; Fan et al. 2017), spoken by around 5000 people in Liuhe Township (六合乡, *Liùhé Xiāng*), Heqing County (鹤庆县, *Hèqìng Xiàn*), Dali Bai Autonomous Prefecture, Yunnan Province, China. Despite the small population, Kua'nsi children are still actively learning and speaking the language, and Kua'nsi is used in most domains in the villages, except in schools, where Mandarin Chinese is the prominent language. Multilingualism is common among Kua'nsi speakers. The major ethnic group in Heqing County is Bai (白族, báizú) and most Kua'nsi people can speak Bai and the local variety of Southwestern Mandarin at different levels. As Mandarin Chinese has become the more socially dominant language, the multilingual pattern has changed. While Kua'nsi elders are bilingual in Kua'nsi and Bai, young Kua'nsi people are more proficient in Mandarin and sometimes with limited knowledge of Bai in addition to Kua'nsi as their first language.

In Kua'nsi, the majority of words are monomorphemic, but there are four ways in which words can be formed: compounding, affixation, tonal inflection, and reduplication. Kua'nsi reduplication shows more complex forms and functions than the other three word-formation strategies.

Reduplication involves a systematic repetition of some components of a word with a semantic or grammatical purpose (Rubino 2005, p. 11). Different from other morphemes, the shape of reduplicative morphemes (reduplicant) depends entirely on the reduplicated base (base), and the base can be an entire word or any subconstituent of a word at different morphological or phonological levels, such as a root, affix, or syllable of root (Inkelas and Downing 2015, p. 504). Below, (1) shows some reduplication of stative verbs in Kua'nsi. As the description of reduplication implies, it is considered a morphological process in this paper, and so, by definition, any syntactic repetitions of words or phrases are excluded from consideration. For example, in Kua'nsi, quantifying phrases that consist of a quantifier and a classifier can be repeated, as in (2), with the quantifying phrase *tɕʰi²¹ gu²¹* 'one chunk'. The repetition of quantifying phrases is not within the word level and thus is not

considered a case of reduplication in Kua'nsi. The difference between morphological reduplication, especially the full reduplication, and syntactic repetition of words in Kua'nsi is that the morphological reduplication cannot be interrupted by other words such as conjunctions $ni^{33}$ 'and' and $dzi^{33}$ 'and (then)', while this is possible for syntactic repetition.

(1)　　$bja^{33}$　　　　　　　　　　'bland'　　　　　$bja^{33}{\sim}bja^{33}$　　　　'very bland'
　　　　$ni^{55}$　　　　　　　　　　'surprised'　　　$ni^{55}{\sim}ni^{55}$　　　　'very surprised'
　　　　$swa^{33}wo^{21}$　　　　　'yellow'　　　　　$swa^{33}wo^{21}{\sim}wo^{21}$　　'(very) yellow'
　　　　$tsa^{33}pi^{21}$　　　　　　'shameless'　　　$tsa^{33}tsa^{33}{\sim}pi^{21}$　　'very shameless'

(2)　$si^{21}t\varepsilon ja^{21}$　　$dzi^{33}$　　**$t\varepsilon^h i^{21}$**　**$gu^{21}$**　　**$t\varepsilon^h i^{21}$**　**$gu^{21}$**　　$su^{21}$　　$dwa^{21}$
　　　fire.wood　　TOP　　one　　CLF:chunk　　one　　CLF:chunk　　COMP　　chop
　　　'(he) chopped the firewood into chunks (or chunk by chunk)'

Formally, reduplication can be categorised as either partial or full, based on whether the reduplicated base is the entire word or a subconstituent of it (Rubino 2005; Inkelas 2014). In full reduplication, the whole word is the reduplicated base. In partial reduplication, only some subpart of the word is duplicated, such as a syllable or a consonant. Cross-linguistically, while full reduplication is accompanied by minimal or no phonological modification, partial reduplication often occurs with concomitant phonological modifications (Inkelas 2014, p. 170). The relation between the reduplicated form and its base raises many theoretical questions. For example, what is the best way to formalise the phonological derivation between the reduplicant and the base? What is the morphological status of reduplication morphemes? Is the process of reduplication phonologically or morphologically driven? Working within different theoretical perspectives, many models of the derivations between the reduplicant and its base have been proposed (Marantz 1982; McCarthy and Prince 1986, 1995; Raimy 2000; Inkelas and Zoll 2005; Saba Kirchner 2010; and many others).

Reduplication can be employed to encode many functions within a language. Three major types of functions can be found across languages. It can be used (a) as inflectional morphology to encode person, number, aspect, and other grammatical categories; (b) as derivational morphology to change part of speech, change the valency of verbs, or create a new word; and (c) without any semantic or grammatical purpose but as a concomitant feature of affixation or a repair for ill-formed phonological or templatic structures (Rubino 2005; Inkelas 2014; Inkelas and Downing 2015). Reduplication is commonly used to denote plurality, tense, aspect, attenuation, intensity, transitivity, reciprocity, and so on (Rubino 2005, p. 19). Some functions of reduplication are often argued to show iconicity, and proposals are put forward to connect more frequent functions of reduplication to more peripheral ones (see Key 1965; Moravcsik 1978; Kiyomi 1995; Regier 1998; Fischer 2011; Li and Ponsford 2018).

The relation between the form and function of reduplication is not always straightforward. It is not uncommon to find that one reduplicant form may be used to encode different functions (one-to-many) or many forms to encode one function (many-to-one). For example, in Xong (Miao-Yao), only full reduplication of verbs is found. It is used to indicate intensity on the property- and state-denoting verbs or repetitive actions on other verbs (Sposato 2021). In the reduplication of other languages, one-to-many or many-to-one relations can be found. In Japhug (Sino-Tibetan), where reduplication shows a one-to-many relation, partial reduplication of the initial syllable of the verb complex can mark 'the protasis of a conditional construction, iterative coincidence, degree incrementation, and totalitative relativisation' (Jacques 2021, p. 71). As we will see, some forms and functions of reduplication show many-to-one relations in Kua'nsi. For example, the function of intensification with adjectives and stative verbs is encoded by several reduplication forms in Kua'nsi.

Despite the high degree of morphological diversity in Sino-Tibetan languages (Chung et al. 2015; Arcodia and Basciano 2020), reduplication is widespread in the family. For ex-

ample, the discussion of reduplication can be found in numerous chapters in the collection of sketch grammars (LaPolla and Thurgood 2017) and more detailed descriptive grammars of Sino-Tibetan languages (e.g., Huang 2004; Coupe 2007; Willis 2007; Daudey 2014; Zemp 2018; Jacques 2021; Konnerth 2022; and many others).

Reduplication is employed to encode many different functions in Sino-Tibetan languages. Apart from those that are common cross-linguistically, some functions of reduplication found in the Sino-Tibetan languages are rare outside the family. For example, some Ngwi languages, including Kua'nsi, use verbal reduplication to form polar questions[1] (Gerner 2013, p. 46; Bradley 2017, p. 914; Donlay 2019, p. 395). This function of reduplication has not been presented in previous typological works on reduplication (e.g., Rubino 2005; Lǐ and Ponsford 2018). This is possibly because such works do not include a balanced sample of Sino-Tibetan languages. For example, the recent research on the predicative function of reduplication, Lǐ and Ponsford (2018), only includes one language from the family, Mandarin Chinese, which does not account for the diverse functions of reduplication, especially within the Tibeto-Burman branch of the family.

This paper presents reduplication in Kua'nsi as a showcase of the diverse forms and functions found in Sino-Tibetan languages. Reduplication in Kua'nsi has three forms: full, partial, and discontinuous. The distribution of reduplication in Kua'nsi is not restricted and is found in the formation of classifiers, verbs, adjectives, and adverbs. Within each syntactic category, reduplication may encode one or more functions. One feature of reduplication in Kua'nsi is that, although it is productive in the language, it is also lexicalised, because the tonal change is only available to some words, but not others, and there is not a rule to predict this. Different reduplication patterns may denote the same function and there are no specific rules to predict the form of reduplication with each word. In Section 2, I briefly introduce the language features related to the reduplication forms in Kua'nsi. The forms and functions of reduplication in Kua'nsi are discussed in Sections 3 and 4.

The data presented in this paper were collected by the author during his fieldwork in Kua'nsi villages. All example sentences are from spontaneous conversations. In the examples where only words are listed, the data are from both the conversation and elicited data. For the investigation of reduplication, a word list was specifically compiled. The speakers were asked to produce the reduplication form of each word on the list. The speakers also provided other common reduplication words that were not listed after a discussion with other elder Kua'nsi speakers.

## 2. Typological Overview of Kua'nsi

In this section, I will present some typological features of Kua'nsi that are important to understand the forms and functions of Kua'nsi reduplication. The first is the low-rising tone /13/, which is a peripheral tone with underived words but is frequently found with reduplication. The second feature is syllable structure and shape. Although the basic reduplicative unit in Kua'nsi is a syllable, some syllables cannot be reduplicated, and others, such as syllables /ʔV/ and /V/, only occur infrequently. I will also show some examples of other word-formation strategies and an overview of the word classes in Kua'nsi.

### 2.1. Tones

Kua'nsi has three primary lexical tones: low-falling /21/, mid-level /33/, and high-level /55/[2], as exemplified by the minimal pair set given in (3) based on the syllable /ta/. Kua'nsi vowels show a distinction between modal ('lax') and laryngealised ('tense') phonation, which is widely attested in other Ngwi languages, such as Nuosu Yi (Edmondson et al. 2017) and Central Lisu (Tabain et al. 2019). The lax vowels can occur with all three tones, while tense vowels can only occur with low-falling and mid-level tones.

(3)          Near minimal quintuplet set of Kua'nsi tones based on the syllable /ta/
    $p\dot{i}^{33}l\dot{i}^{33}ta^{21}$          'sour'
    $ta^{33}$          'take, hold'
    $a^{55}ta^{55}\dot{i}^{33}$          'uncle (father's older brother)'
    $ta^{21}$          'step (on)'
    $t\underline{a}^{33}$          'hold in arms, hug'

Besides these three tones, there are two other contour tones, low-rising /13/ and high-falling /51/. /51/ occurs when a word-final syllable with a /55/ tone is contracted with the following syllable consisting of a single vowel with /21/ tone; for example, $xu^{55}$ 'PFV' + $=a^{21}$ 'SFM' → $xwa^{51}$ 'PFV.SFM'. However, the syllable contraction is not systematic. The outcome of this tonal fusion is sometimes /21/ or close to /21/; for example, $\eta u^{55}$ '1SG' + $=a^{21}$ 'CM' → $\eta wa^{21}$ '1SG.CM'. The other contour tone /13/ is found with only one word[3], the augmentative particle $dwa^{13}$ (free variation with $lwa^{13}$) 'AUG, very'. However, it is often found as the initial syllable in reduplication, which is discussed in Section 3.

### 2.2. Syllable Structure

Kua'nsi syllables can be N, CV, CVV, CCV, or CCCV, as shown in (4). Note that N represents a syllabic nasal or a vowel. Syllables in Kua'nsi are all open, i.e., there is no syllable coda. The most common syllable in Kua'nsi is CV[4]. Kua'nsi words are mostly monosyllabic or disyllabic. Thus, the structure of Kua'nsi words is often CV or CVCV. The syllable CVV occurs only in the context of the syllable contraction, e.g., $t\varepsilon^{55}$ 'return' + $=a^{21}$ → $t\varepsilon a^{21}$ 'return.SFM'.

(4)          N          $n^{21}$          'not'
    CV          $gu^{21}$          'chew'
    CVV          $t\varepsilon a^{21}$          'return.SFM'
    CCV          $p^{h}ja^{55}$          'cloth'
    CCCV          $ʔmja^{21}kwa^{21}$          'donkey'

In Kua'nsi reduplication, the basic reduplicative unit is a syllable, but the syllable structures in (4) have different distributions in reduplication. First, the syllables /ʔV/ and /V/ are restricted. This is possibly because they have a relatively low overall frequency in the Kua'nsi lexicon. Second, two types of syllables are never found in reduplication as the reduplicant and the base. The first is /CVV/ and the other is the syllabic nasal. The syllable structure /CVV/ never occurs in the reduplication, because this type of syllable results from syllable contraction in speech. Thus, it is not part of any morphological word in Kua'nsi. The reason syllabic nasals are not reduplicated is because of their distributions in different word classes. The syllabic nasals only occur when the onset of the following syllable is a homorganic obstruent and usually in nouns; for example, $n^{21}ts^{h}\varepsilon^{55}$ 'hair', $m^{21}pi^{21}$ 'comb', $\eta^{21}ka^{55}$ 'head'. However, I did not find any noun that can be reduplicated in Kua'nsi. With adjectives and verbs, the syllabic nasals usually occur as the negative prefix. For example, $g\varepsilon^{21}$ 'give' vs. $\eta^{21}-g\varepsilon^{21}$ 'not give'; $mja^{33}$ 'see' vs. $m^{21}-mja^{33}$ 'not see'. When the syllabic nasals function as a negative marker with adjectives and verbs, they are never reduplicated[5].

### 2.3. Word-Formation Strategies

The majority of words in Kua'nsi are monomorphemic. However, there are still several ways in which a morphologically complex word can be formed besides reduplication: compounding, affixation, and tonal inflection, as shown in (5).

(5) Word-formation strategies besides reduplication in Kua'nsi
　　a.　Compounding: two independent words are combined to form a single word. For example, $ʔa^{33}ŋ^{21}ka^{55}la^{55}mu^{21}$ + $ʔa^{33}zu^{21}$ sky + bird 'wild goose'; $ʔa^{55}na^{55}$ + $bɡ^{33}$ water + lock 'water well'.
　　b.　Affixation: Kua'nsi has several affixes. For example, the diminutive suffix $-zu^{21}$ can be attached to nouns denoting human, animate, and inanimate references: $ʔa^{33}ha^{21}$-$zu^{21}$ child-DIMN 'small child'; $ɣa^{21}$-$zu^{21}$ pig-DIMN 'small pig'; $ʔlo^{21}dzi^{21}$-$zu^{21}$ stone-DIMN 'small stone'.
　　c.　Tonal inflection: the tonal differences can encode different word forms, but this is not productive in Kua'nsi. It is only used to inflect singular possessive pronouns from singular personal pronouns[6]. For example, $ŋu^{55}$ '1SG' → $ŋu^{21}$ '1SG.POSS'; $ɲi^{55}$ '2SG' → $ɲi^{21}$ '2SG.POSS'; $i^{33}$ '3SG' → $i^{21}$ '3SG.POSS'.

*2.4. Word Classes*

　　Like other Ngwi languages, establishing word classes in Kua'nsi is primarily based on syntactic criteria, including (i) the distribution of the word, such as the clausal position at which the word can occur, and (ii) the syntactic function, such as its ability to function as an argument, modifier, or predicate. Although Kua'nsi is not rich in morphology, there are several morphological derivations that are helpful to differentiate major word classes. For example, (active) verbs can take the imperative prefix $a^{33}$-; most members of nominal categories can be marked by the case marking clitic $=a^{21}$.[7] Based on these morphosyntactic criteria, Kua'nsi distinguishes three major word classes, nouns, verbs, and adjectives, as well as other word classes such as numerals, classifiers, adverbs, conjunctions, and postverbal particles. The major word classes serve as a reference point for distinguishing other word classes in Kua'nsi, as it is possible to use them to refer to the relative position of other word classes in the clause. They are also the word classes that can undergo the process of reduplication, thus it is important to illustrate the primary morphosyntactic features of these word classes.

　　Nouns are one of the large open word classes in Kua'nsi. They refer to a person, animal, or thing, such as artefacts, instruments, flora and fauna, body parts, and so on. Kua'nsi nouns do not inflect for number, gender, or their grammatical categories, but they can take the diminutive suffix $-zu^{21}$ and the case marking clitic $=a^{21}$. In noun phrases, the nouns can be followed by a demonstrative, quantifying phrase (quantifier + classifier) and article in the listed order. Examples (6) and (7) show two noun phrases within which the nouns occur at the beginning and are followed by words from other nominal categories.

(6)　[$a^{33}ha^{21}$-$zu^{21}$　　　　$xu^{21}$　　　　$ɲi^{21}$　　　　$mu^{33}$]$_{NP}$　　　$mɡ^{33}$　　　　$xu^{55}=a^{21}$
　　　child-DIMN　　　　DIST　　　　two　　　　CLF:GEN　　　sleep　　　　PFV=SFM
　　　'those two children fell asleep'

(7)　[$zi^{21}mu^{33}$　　　　$tɕju^{33}$　　　　$tɕ^hi^{21}$　　　$tsɡ^{33}$]$_{NP}$
　　　river　　　　DIST.low　　　one　　　CLF:river
　　　'that river (down there)'

　　Verbs can be distinguished from nouns based on many aspects of their behaviours. Most importantly, they differ in their syntactic functions. Nouns function as arguments and verbs as predicates in the clause. Morphosyntactically, verbs can take the negative prefix $N^{21}$-, which is realised as a homorganic nasal with the onset of the following syllable, and they can be followed by a series of postverbal particles that denote aspect, associated motions, applicative, and causative. Kua'nsi verbs can also combine to form serial verb construction, while members of other word classes are usually not able to occur juxtapositionally. Example (8) shows the verb $ɲi^{55}$ occurs in the clause with the imperative prefix $a^{33}$- and is followed by the causative particle $tsi^{55}$. In (9), a series of verbs occur together in the predicate complex indicated by the brackets and the final verb in the series takes the

negative prefix.

(8)     $xwa^{33}$     $[a^{33}\text{-}\textipa{ɲ}i^{55}$     $tsi^{55}]_{\text{PRED}}$
        DIST.PLC     IMP-sit     CAUS
        '(he made him) sit there'

(9)     $i^{33}$     $[\eta a^{33}$     $\textipa{ɕ}ju^{21}$     $n^{21}\text{-}u^{33}\text{=}a^{21}]_{\text{PRED}}$
        3SG     bite     eat     NEG-get=SFM
        'it (the dog) did not succeed in biting and eating (the cat)'

Adjectives can also occur as predicate and share some morphosyntactic features with verbs, such as hosting the negative prefix $N^{21}\text{-}$ and being followed by some but not all postverbal particles that can follow verbs. The major difference between adjectives and verbs is that, as modifiers to nouns, adjectives can occur directly after or before the head noun, as shown in (10) and (11), while it is not possible for verbs to directly occur as modifiers within noun phrases.

(10)     **$x\textipa{ɥ}\textipa{ɛ}^{21}$**     $ts^ha^{33}$     $t\textipa{ɕ}^hju^{55}$     $ka^{21}$     $lu^{33}$
        bad     person     one.CLF:GEN     arrive     come
        'here comes a bad person'

(11)     $\text{ʔ}\underline{u}^{33}nu^{21}\text{-}\textipa{z}u^{21}$     **$bo^{21}lo^{33}\text{~}lo^{33}$**     $a^{21}$     $t\textipa{ɕ}^hju^{55}$
        dog-DIMN     colorful~RDP     PROX     one.CLF:GEN
        'this colourful little dog'

Another distinction between adjectives and verbs, especially stative verbs, is that stative verbs can trigger the case marking on the S argument, while adjectives cannot. For example, in (12), the stative verb $dzja^{33}k\textipa{ɘ}^{21}$ '(feel) cold' occurs in the predicate, and the S argument is marked with the case marking clitic $\text{=}a^{21}$.

(12)     $i^{33}\text{=}a^{21}$     $dzja^{33}k\textipa{ɘ}^{21}$     $dwa^{13}$
        3SG=CM     feel.cold     AUG
        'he felt very cold'

In addition to nouns, verbs, and adjectives, another word class related to the process of reduplication is adverbs. Adverbs provide additional information about the event denoted by the clause, such as time, manner, location, degree, and so on. Kua'nsi adverbs cannot occur within predicate complex and noun phrases. Although adverbs can occur in multiple places within the clause, they tend to occur in the preverbal position, as in (13), where the adverb $do^{21}$ occurs before the verb $\textipa{ɲ}i^{21}p^hi^{33}$ 'jealous'.

(13)     $i^{33}$     $do^{21}$     $\textipa{ɲ}i^{21}p^hi^{33}\text{=}a^{21}$
        3SG     very     jealous=SFM
        'he is very jealous'

The preverbal position is also where adverbial expressions often occur. In Kua'nsi, the nominalised clause with the manner nominaliser $su^{21}$ can function as an adverbial subordinated clause and often occurs before the predicate complex of the matrix clause. For example, in (14), the preverbal nominalised clause in brackets occurs as an adverbial clause to specify the manner of the main event of walking.

(14)     $[ma^{33}\textipa{ɲ}\textipa{ɘ}^{21}$     $la^{21}$     $di^{33}$     $su^{21}]_{\text{NML}}$     $zwa^{21}$
        bag     also     carry     NMLZ.manner     walk
        'he is walking while carrying a bag'

### 3. The Forms of Reduplication in Kua'nsi

Cross-linguistically, the form of reduplication can be examined broadly in two ways: (a) full vs. partial: whether the whole or part of the word is reduplicated; and (b) simple vs. complex: whether or not the reduplicant is the exact copy of the reduplicated base (Rubino 2005; Inkelas and Downing 2015). The categorisation of simple vs. complex reduplication is a cover term for the great cross-linguistic diversity of phonological variation between a reduplicant and its base.

The reduplicant can be placed in different positions relative to the base, as a prefix, infix, or suffix (Rubino 2005; Inkelas and Downing 2015). However, it is not always the case that the reduplicant has to occur next to its base. The reduplicant and the base can be interrupted by other morphological material, and this type of reduplication is termed discontinuous reduplication, which is a non-canonical type of reduplication (Stolz 2018; Mattiola and Masini 2022).

Kua'nsi reduplication can be categorised into three types: full, partial, and discontinuous. Table 1 lists the different patterns and examples of reduplication in Kua'nsi. In all three types of reduplication, the basic unit of Kua'nsi reduplication is always a syllable. It is either one syllable of a word or a whole word that is reduplicated. A single consonant or vowel cannot be the base of reduplication, and thus not all syllables can be reduplicated. For ease of description, the syllable of the base word that is reduplicated will be referred to as the base syllable, and the resulting reduplicated syllable will be referred to as the reduplicated syllable.

**Table 1.** Reduplication patterns in Kua'nsi.

| Pattern | | Example |
|---|---|---|
| Full reduplication | | $ni^{55}$ 'surprised' → $ni^{55}$~$ni^{55}$ 'very surprised' |
| Partial reduplication | Initial syllable | $\epsilon i^{33}k\vartheta^{55}$ 'jealous' → $\epsilon i^{33}$~$\epsilon i^{33}k\vartheta^{55}$ 'very jealous' |
| | Final syllable | $si^{21}ni^{55}$ 'red' → $si^{21}ni^{55}$~$ni^{55}$ 'very red' |
| Discontinuous reduplication | | $si^{21}ni^{55}$ 'red' → $si^{21}ts^hi^{33}$~$si^{21}ni^{55}$ 'very red' |
| | | $bo^{21}$ 'bright' → $bo^{21}$~$li^{55}bo^{21}$ 'brightly' |

Note: the symbol '~' indicates the boundary between the reduplicant and the base.

Kua'nsi full and partial reduplication is not simple, as tonal modification may occur in both types. In discontinuous reduplication, an additional monosyllabic segment occurs between the reduplicant and the base. There is no phonological modification of the reduplicated syllable in the discontinuous reduplication.

#### 3.1. Full and Partial Reduplication

As expected, full reduplication in Kua'nsi copies the whole word. When the base is a polysyllabic word, the reduplicant does not involve any kind of phonological modification. However, when the base is a monosyllabic word, the tone of some reduplicant syllables is /13/, while the reduplicants of other monosyllabic and all polysyllabic words retain the tone of the base syllable. A change in tone is also attested in the partial reduplication of some words, and this will be further discussed in Section 3.3. The distribution of full reduplication is only found with some adverbs and monosyllabic words, as shown in (15). Other polysyllabic words including nouns, adjectives, and verbs do not occur in full reduplication.

(15)  $ga^{21}$              'broken'      $ga^{21}$~$ga^{21}$                    'very broken'
      $vi^{33}$              'high'        $vi^{33}$~$vi^{33}$                    'very high'
      $ts^h\vartheta^{33}$   'greedy'      $ts^h\vartheta^{13}$~$ts^h\vartheta^{21}$  'very greedy'
      $mo^{21}t\varepsilon^hi^{33}$  'quietly'  $mo^{21}t\varepsilon^hi^{33}$~$mo^{21}t\varepsilon^hi^{33}$  'very quietly'
      $lu^{33}swa^{21}$      'slowly'      $lu^{33}swa^{21}$~$lu^{33}swa^{21}$    'very slowly; gradually'

Partial reduplication is found with polysyllabic words in Kua'nsi, and it involves the doubling of either the initial or the final syllable of a word. The syllables in the middle of a polysyllabic word, such as the middle syllable of a trisyllabic word, are never found to be duplicated. Some examples of partial reduplication of initial syllables are shown in (16) and of final syllables in (17). Partial reduplication can be found with all word classes whose members can be reduplicated.

(16)　　$vi^{33}\sim vi^{33}dzi^{21}$　　　　　　RDP~dark　　　　　'very dark'
　　　　$\varepsilon i^{33}\sim\varepsilon i^{33}k\vartheta^{55}$　　　　　　RDP~jealous　　　'very jealous'
　　　　$nu^{33}\sim nu^{33}u^{33}\textbardbl nu^{33}$　　　RDP~annoying　　'very annoying, irritating'
　　　　$bo^{13}\sim bo^{21}lo^{21}$　　　　　RDP~colourful　　'very colourful'
　　　　$k^ha^{13}\sim k^ha^{21}mo^{33}$　　　RDP~regretful　　'very regretful'

(17)　　$na^{21}kwa^{21}\sim kwa^{21}$　　　　black~RDP　　　'very black'
　　　　$si^{21}\textbardbl ni^{55}\sim\textbardbl ni^{55}$　　　　　red~RDP　　　　'very red'
　　　　$u^{55}ts^hu^{55}\sim ts^hu^{55}$　　　　hot~RDP　　　　'very hot'
　　　　$\vartheta^{33}xu^{21}\sim xu^{21}$　　　　　ugly~RDP　　　'very ugly'
　　　　$gwa^{33}t\varepsilon^hja^{33}\sim t\varepsilon^hja^{33}$　　slim~RDP　　　'very slim'

While partial reduplication of the final syllable does not have any phonological modification, partial reduplication of the initial syllable may involve a tonal modification on both reduplicated and base syllables. This will be discussed in Section 3.3.

### 3.2. Discontinuous Reduplication

There are two types of discontinuous reduplication in Kua'nsi. The first one is found with words denoting colours. In the reduplication of these words, the first syllable is reduplicated and followed by an additional segment $ts^hi^{33}$, which is not present in the base word. The resulting syllables are attached to the left of the base, as shown in (18). Note that these colour words can also undergo the partial reduplication of final syllables; for example, $na^{21}kwa^{33} \rightarrow na^{21}kwa^{33}\sim kwa^{33}$ 'very dark'; $si^{21}\textbardbl ni^{55}$ 'red' $\rightarrow si^{21}\textbardbl ni^{55}\sim\textbardbl ni^{55}$ 'very red'.

(18)　　$na^{21}ts^hi^{33}\sim na^{21}kwa^{33}$　　RDP~black　　　'very black'
　　　　$si^{21}ts^hi^{33}\sim si^{21}\textbardbl ni^{55}$　　　RDP~red　　　　'very red'
　　　　$si^{21}ts^hi^{33}\sim si^{21}wo^{21}$　　　RDP~yellow　　'very yellow'

The second type of discontinuous reduplication is found with some monosyllabic words. In this form of reduplication, an extra syllable $li^{55}$ is inserted between the base and reduplicated syllables, as shown in (19).

(19)　　$bo\underline{o}^{21}li^{55}\sim bo\underline{o}^{21}$　　　bright~RDP　　'bright'
　　　　$wa^{21}li^{55}\sim wa^{21}$　　　big~RDP　　　'largely'
　　　　$lo^{21}li^{55}\sim lo^{21}$　　　　clean~RDP　　'clean'
　　　　$tju^{21}li^{55}\sim tju^{21}$　　　thick~RDP　　'thick'
　　　　$t\varepsilon ja^{21}li^{33}\sim t\varepsilon ja^{21}$　　　lively~RDP　　'lively'

Some of these monosyllabic bases in (19), such as $bo\underline{o}^{21}$ and $t\varepsilon ja^{21}$, cannot occur alone and must be in the reduplicated form. Alternatively, these reduplicated forms could be analysed synchronically as words with repeated syllables. However, I treat them as the reduplicated forms, because they show the same word structure as some monosyllabic adjectives that can occur in reduplicated form and independently. For example, the adjective $wa^{21}$ can occur in this form of reduplication, as in (19), and it can also be used as an independent word, as in (20).

(20)　　$nu^{21}$　　　$\textbardbl nu^{21}$　　$xu^{21}$　　$t\varepsilon^hju^{55}$　　　$dzi^{33}$　　**$wa^{21}$**　$p^hi^{33}fi^{21}$
　　　　2PL.POSS　cattle　　that　　one.CLF:GEN　TOP　**big**　AUG
　　　　'your cow is very big'

It is often impossible to know the nature of the interposing elements in discontinuous reduplication in many languages, but there are cases where the interposing elements also have other functions in the languages (Mattiola and Masini 2022). In Kua'nsi, the segment $ts^hi^{33}$ in the discontinuous reduplication of colour words cannot be interpreted, as it is not used anywhere else in Kua'nsi. However, the interposing element $li^{55}$ in the second type of discontinuous reduplication has the same form as the appearance nominaliser $li^{55}$. The nominaliser can occur after nouns, adjectives, and verbs, which denote the appearance of a referent, such as colour, length, size, and so on. The derived nominal refers to a referent that has the same kind of characteristics, as shown in (21). Although I do not have strong evidence to show a link between the nominaliser $li^{55}$ and the interposing element in the discontinuous reduplication, it is possible that these two elements are related. This is because both of them occur after property terms and they share the same form.

(21)     a.      $wa^{21}$                       $li^{55}$
                 big                             NMLZ
                 'the big one'
         b.      $swa^{33}wo^{21}$               $li^{55}$
                 yellow                          NMLZ
                 'the yellow one'

*3.3. Tonal Modification in Full and Partial Reduplication*

Kua'nsi full and partial reduplication may involve additional tonal modification. The tonal modification occurs with the partial reduplication of initial syllables and the full reduplication of monosyllabic words. The tonal modification is the same in both forms of reduplication. The tone on the initial syllable changes from /21/ or /33/ to /13/. As mentioned in Section 2.1, /13/ tone only occurs with the augmentative particle in Kua'nsi. Apart from this, /13/ is unique to the reduplication, but it does not always appear with the reduplication.

In the initial syllable reduplication, if the tone of the base syllable is not the high-level tone /55/, then the tones of the reduplicated and base syllables will be /13/ and /21/, respectively, as shown in (22). A prosodic feature associated with the tone change is that the length of reduplicated syllable is longer than the base syllable.

(22)     $\varepsilon\gamma\varepsilon^{21}k^h\vartheta^{33}$          $\varepsilon\gamma\varepsilon^{13}\!\sim\!\varepsilon\gamma\varepsilon^{21}k^h\vartheta^{33}$          'very angry'
         $hi^{21}mi^{21}$          $hi^{13}\!\sim\!hi^{21}mi^{21}$          'very hungry'
         $tu^{21}si^{33}$          $tu^{13}\!\sim\!tu^{21}si^{33}$          'very sensible'
         $t\varepsilon ja^{33}k\vartheta^{21}$          $t\varepsilon ja^{13}\!\sim\!t\varepsilon ja^{21}k\vartheta^{21}$          'very cold'

Figure 1 shows the pitch contours of $hi^{21}mi^{21}$ and its reduplicated form $hi^{13}hi^{21}mi^{21}$ 'very hungry'. The pitch of both syllables of $hi^{21}mi^{21}$ are falling, but the first syllable of its reduplicated form shows a rising pitch followed by a falling pitch on the second syllable. It clearly shows that the tones of reduplicated syllables do not match that of the base syllable in initial syllable reduplication.

Note that this tone change is not found with all initial syllable reduplications. As mentioned, if the tone of the base is high-level tone /55/, there is no tone change involved in reduplication. Some base syllables with other tones may also retain the tones in reduplicated form. For example, in $\varepsilon i^{33}\varepsilon i^{33}\vartheta^{33}$ 'very jealous (base: $\varepsilon i^{33}\vartheta^{33}$)', neither the tone nor the length of reduplicated syllable is different from the base syllable.

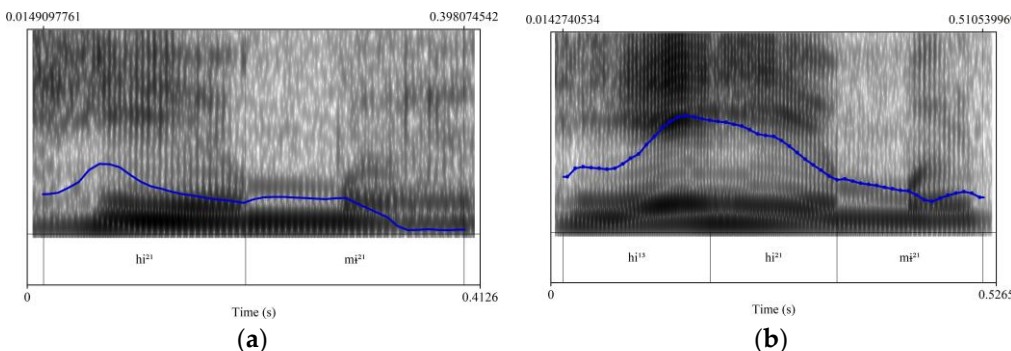

**Figure 1.** Pitch contours of hi²¹mi²¹ 'hungry' in (**a**) and its reduplicated form. hi¹³hi²¹mi²¹ 'very hungry' in (**b**).

The same tone change is also found with some of the reduplications of monosyllabic words, as shown in (23). This points to two different categories of the reduplication of monosyllabic words: one group of words involves tone change in reduplication, and the other has an exact copy of the base. Note that the monosyllabic words with the /55/ tone also do not show any tonal modification in reduplication.

(23)    *wa²¹*          *wa¹³~wa²¹*          big~RDP            'very big'
        *dzju²¹*        *dzju¹³~dzju²¹*      frightened~RDP     'very frightened'
        *ka³³*          *ka¹³~ka²¹*          most~RDP           'the most'

This raises an interesting question about the underlying reduplication mechanism with monosyllabic words. That is, it raises the question of whether the reduplication involving tonal modification and the exact reduplication of monosyllabic words are indeed the same process of reduplication. It might be possible that monosyllabic words rather show the same reduplication process as initial syllable reduplication, as the tone change can occur with some monosyllabic words. The monosyllabic words are associated with full reduplication, just because of the number of syllables in these words. Underlying, there might be no difference in the mechanisms of those reduplication patterns.

### 3.4. Unpredictability and Variation of Reduplication

The form of reduplication of each word in Kua'nsi is not always predictable, but rather lexicalised. Words within the same syntactic category will take different forms of reduplication for the same function, and the tone change with some but not all reduplicated forms makes it more unpredictable. That is, there are no rules that predict which reduplication patterns occur with each word. Words taking one form of reduplication do not show any semantic or syntactic difference from words taking another form of reduplication. The exceptions to this unpredictability are the discontinuous reduplication, which is only available with colour words, and the base syllables whose tone is /55/.

The form of reduplication is also not straightforward for monosyllabic words. The basic unit of reduplication in Kua'nsi is the syllable, and no subconstituent of the syllable, such as onset or coda, can undergo reduplication. This means that a monosyllabic word always doubles its only syllable in reduplication. However, the possibility of tone change means that the reduplication of monosyllabic words is not always an exact copy of its base.

The unpredictable reduplication forms are not limited to more frequently used words. As the partial and full reduplication patterns are productive, infrequent and borrowed words can also occur with those unpredictable forms of reduplication. For example, *xo²¹gu³³* 'bossy' → *xo¹³xo²¹gu³³* 'very bossy'; *tseŋ²¹tɕʰi²¹* 'tidy' (borrowed from Mandarin Chinese 整齐 *zhěngqí* 'tidy') → *tseŋ¹³tseŋ²¹tɕʰi²¹* 'very tidy'.

These unpredicted forms seem to more often occur with the intensification function with adjectives and stative verbs, and less so when the reduplication is used to form polar questions. As suggested by a reviewer, this unbalanced distribution of the tonal combi-

nation (/13/ + /21/) in different functions of reduplication may imply that the unexpected tonal combination highlights certain functions of reduplication, thus providing a prosodic emphasis alongside the morphological emphasis. This could be a possible explanation for the occurrence of the tonal combination. However, the tonal combination does not occur every time with reduplication when the function is to intensify the meaning. This tonal combination is also banned when the original tone is /55/. Thus, it is still unclear why the original tones are maintained sometimes, but change at other times, even though the reduplication denotes the same function.

Although the form of reduplication is lexicalised, the association between each word and its reduplicated form is not always fixed, and I have observed some variations in the reduplicated form of the same word. For example, both types of partial reduplication are sometimes possible for the same word. In (24), the final syllable of the word $k\vartheta^{55}t^h\vartheta^{21}$ 'fat' is reduplicated, while in (25), it is the first syllable of the same word that is reduplicated. These two examples were given by two different speakers, and a third speaker rejected the use of initial syllable reduplication in (25). This might suggest dialectal variation in reduplicated forms (these three speakers come from different villages), or this could be some degree of speaker variation that is idiolectal.

(24)    $du^{21}$        $k\vartheta^{55}t^h\vartheta^{21}{\sim}t^h\vartheta^{21}$     $su^{21}$        $t\varepsilon^hju^{55}$     $ka^{21}$       $lu^{33}$
       more       fat~RDP        COMP     one.CLF:GEN   arrive      come
       'here comes a fatter man'

(25)    $a^{33}$-$na^{21}$    $k\vartheta^{55}{\sim}k\vartheta^{55}t^h\vartheta^{21}$    $dwa^{13}$
       IMP-look     RDP~fat         AUG
       'look! (it is) very fat'

## 4. The Function of Reduplication

Cross-linguistically, reduplication can encode many functions and is used with inflectional and derivational meanings, as well as with no apparent semantic or syntactic effect (Inkelas 2014). In Kua'nsi, reduplication is used to indicate plurality, indicate inclusivity, intensify the meaning, change word class, form polar questions, indicate totalitativity and form new words together with noun incorporation. It is also used as a concomitant feature without adding any extra semantic component to the overall meaning of a sentence: it is obligatory in the nominalisation of some words. While other functions are common cross-linguistically, two functions, forming polar questions and forming new words together with noun incorporation, are possibly rare across languages.

While partial and full reduplication patterns can denote all of these functions, those two types of discontinuous reduplication patterns are limited in some functions. The discontinuous reduplication of colour words only denotes the function of intensification of meaning, and the discontinuous reduplication in which the segment $li^{55}$ is inserted is only used to change the word class. Note that, in each function, full and partial reduplication patterns can show the tone change.

Some functions of reduplication in Sino-Tibetan languages seem to be restricted to some particular groups. It seems that using reduplication to form polar questions is only found in Ngwi languages in the Sino-Tibetan family. It is commonly found in Qiangic languages that reduplication is used to indicate reciprocity (Huang 2004, p. 200; Daudey 2014, p. 116; Honkasalo 2019, p. 269; Jacques 2021, p. 903), but this use of reduplication seems to be absent in Ngwi languages.

### 4.1. Plurality and Inclusivity

Reduplication can be used to indicate plurality and inclusivity with classifiers in Kua'nsi. Classifiers are the only nominal category that allows reduplication. Compared with other categories allowing reduplication, reduplication with classifiers is peripheral. Speakers do not use this frequently in natural speech, and not every classifier can be reduplicated.

Reduplication with classifiers can denote plurality, as shown in (26) and (27). Note that this is not a primary realisation of plurality in Kua'nsi, and nouns and words of other nominal categories do not take reduplication to denote plurality. As also shown in these two examples, when the classifier is reduplicated, the lexical quantifier $a^{21}nu^{33}$ 'many' will precede it, rather than a numeral quantifier.

(26) $a^{21}nu^{33}$   $ta^{55}{\sim}ta^{55}$
   many   CLF:place~RDP
   'many places'

(27) $jo^{21}swa^{33}$  $a^{21}nu^{33}$  $mu^{33}{\sim}mu^{33}$  $dza^{55}$
   doctor   many   CLF:GEN~RDP  exist
   'there are many doctors'

Reduplication with classifiers can also indicate the meaning of inclusivity, i.e., 'all the members in a group'. For example, $hi^{33}mu^{33}$ 'CLF:family' is reduplicated as $hi^{33}hi^{33}mu^{33}$ to refer to 'the whole family' in (28). In this example, the classifier is preceded by the lexical quantifier $u^{21}$ 'all'.

(28) $i^{21}$  $ni^{21}$  $u^{21}$  $\boldsymbol{hi^{33}{\sim}hi^{33}mu^{21}}$  $\varepsilon i^{13}{\sim}\varepsilon i^{21}d\varphi^{33}$  $lwa^{13}$
   3PL  ART  all  RDP~CLF:family  RDP~generous  AUG
   'their whole family is very generous'

### 4.2. Intensification of Meaning

For stative verbs, adjectives, and adverbs, the function of reduplication is often to intensify or emphasise the meaning. If a reduplicated adjective or stative verb occurs in the predicate complex with this function, one of the augmentative particles, $lwa^{13}/dwa^{13}/pi^{33}fi^{21}$, which all mean 'very', is required following the reduplicated word; otherwise, Kua'nsi speakers will consider it ungrammatical. Below, (29) shows an example of a reduplicated stative verb and (30) an example of a reduplicated adverb. Note that different forms of reduplication do not convey different degrees of intensity or emphasis.

(29) $i^{33}{=}a^{21}$  $\boldsymbol{u^{55}ts^hu^{55}{\sim}ts^hu^{55}}$  $*(lwa^{13})$  $b\varphi^{33}d\varphi^{21}$  $a^{33}nu^{21}?$
   3SG=CM  hot~RDP  AUG  INFR  ADDR.ASYM
   'it seems that he feels very hot, isn't it?'

(30) $t\varepsilon i^{21}zu^{21}$ $t\varepsilon ju^{33}$ $t\varepsilon^hju^{55}$ $dzi^{33}$ $\boldsymbol{ka^{13}{\sim}ka^{21}}$ $ni^{21}$ $li^{33}$ $\eta\varphi^{55}{=}a^{21}$
   bowl  DIST  one.CLF:GEN TOP most~RDP low NMLS COP=SFM
   'that bowl is the lowest one'

### 4.3. Formation of Polar Question

The reduplication of verbs and adjectives in the predicate complex can be used to form polar questions, as shown in (31)–(33). This function of reduplication is also found in some other Ngwi languages. Not every verb in the predicate complex can be reduplicated to denote this function, but specifically, it is the reduplication of the final syllable of the verbs in the predicate complex that derives a polar question. This can be clearly shown by examples with a series of verbs. In Kua'nsi, verbs can be concatenated in the predicate complex, as in (33), where three verbs, $gu^{21}$ 'live', $li^{33}$ 'come', and $no^{33}$ 'want', occur together. To form a polar question, the final syllable of this series is reduplicated, and thus the auxiliary verb $no^{33}$ is reduplicated. Note that particles in the predicate complex cannot be reduplicated for any purpose in Kua'nsi.

(31) $t\varepsilon ju^{33}$ $la^{21}$ $ga^{21}dzu^{33}$ $t\varepsilon^hju^{55}$ $dza^{55},$ $ni^{55}$ $ts\varphi^{21}{\sim}ts\varphi^{21}$ $la^{33}?$
   DIST.low just bridge one.CLF:GEN exist 2SG remember~RDP Q
   'do you remember there is a bridge?'

(32)　$p^hi^{21}t\varepsilon^hju^{21}$　　$dzi^{33}$　　$si^{21}\eta i^{55}{\sim}\eta i^{55}$　　$su^{21}$　　$t\varepsilon^hju^{55}$　　**$\eta\vartheta^{55}{\sim}\eta\vartheta^{55}{=}a^{21}$?**
　　　　ball　　　　　TOP　　　red~RDP　　　COMP　　one.CLF:GEN　COP~RDP=SFM
　　　'is the ball red?'

(33)　$\eta i^{55}$　　　　　$gu^{21}$　　　$li^{33}$　　　**$\eta o^{33}{\sim}\eta o^{33}$?**
　　　　2SG　　　　　live　　　come　　　want~RDP
　　　'do you want to live there?'

Reduplication is analysed as a morphological process, and the functions described so far also have scope over just the reduplicative base at the word level. However, as for the formation of polar questions, the scope of reduplication is clearly a whole proposition. This means that, although the reduplication here is at the word level, its function is at the clause level.

Another use of reduplication with verbs is to show the inference or uncertainty of speakers. As it is common that interrogative forms are used in declarative clauses to show inference or uncertainty, this function of reduplication can be interpreted as an extended function of polar questions formed with reduplication. Below, (34) is taken from a conversation where the speakers are describing the appearance of a character in a picture. One of the speakers is unsure whether the character is a human or something like a leaf, so she uses the reduplication of the verb $zu^{21}$ 'similar' to show that she is not sure about the description. In (35), the speaker is unsure about the event because of unforeseen factors and the auxiliary verb is reduplicated to show the uncertainty.

(34)　$ts^ha^{33}$　**$zu^{21}{\sim}zu^{21}$**　$si^{21}pi^{33}$　**$zu^{21}{\sim}zu^{21}$**　$t\varepsilon^hju^{55}$　$ka^{21}$　$lu^{33}$
　　　　human　similar~RDP　leaf　　similar~RDP　one.CLF:GEN　arrive　come
　　　'here comes someone or something like a leaf'

(35)　$tsa^{21}$　　　　　　$tsi^{55}$　　　　**$du^{33}{\sim}du^{33}$**　　$m^{21}{-}mja^{21}$
　　　　cook　　　　　　　CAUS　　　　able.to~RDP　　　NEG-see
　　　'I am not sure if we are allowed to cook'

### 4.4. Change of Word Class

Another function of reduplication in Kua'nsi is to derive a word with a different part of speech from the original word. This function is often found with the discontinuous reduplication pattern in which the segment $li^{55}$ is inserted, as in (10), and the adjectives and stative verbs become adverbs. This function is not widely attested with full and partial reduplication patterns and only a few examples are found. For example, the adjective $dzwa^{21}$ 'good, beautiful' in (36) occurs in the predicate complex. It can be reduplicated to become an adverb meaning 'properly', as in (37), and occurs before the predicate complex as adverbs, not within the predicate complex.

(36)　$ts^ha^{33}m\vartheta^{33}$　　　　　　**$dzwa^{21}$**　　　　　　$lwa^{13}$
　　　　life　　　　　　　　　　　good　　　　　　　　AUG
　　　'(their) life was very good'

(37)　$a^{21}$　$t\varepsilon^hju^{55}$　$na^{21}$　**$dzwa^{21}{\sim}dzwa^{21}$**　$\eta^{21}{-}\eta i^{55}$　$tu^{21}$　$la^{21}$
　　　PROX　one.CLF:GEN　ART　good~RDP　　　NEG-sit　PRF　ECLM
　　　'this person does not sit properly'

### 4.5. Noun Incorporation

Reduplication in Kua'nsi is also involved in noun incorporation, a word-formation process that results in noun–verb compounds (Mithun 1984; Wang 2022). The noun incorporation involving reduplication in Kua'nsi has a specific construction and meaning. The nominal usually refers to a body part, and the verb denotes the state or action related to the body part. These terms cannot be used alone and they have to be incorporated. In such a construction, the verb is reduplicated as shown in (38). The resulting word can be used to describe how a referent looks. These words can only be used predicatively as matrix verbs

and they cannot modify nouns directly, as adjectives are able to. This construction is not prevalent or productive in Kua'nsi, but there are a few of them.

(38)　　　　a.　　　　$lu^{33}$-$lja^{21}$~$lja^{21}$
　　　　　　　　　　　tongue-lick~RDP
　　　　　　　　　　　'with tongue hanging down'
　　　　　　b.　　　　$s\vartheta^{21}$-$pwa^{33}$~$pwa^{33}$
　　　　　　　　　　　tooth-crooked~RDP
　　　　　　　　　　　'crooked tooth'
　　　　　　c.　　　　$mja^{33}$-$wo^{21}$~$wo^{21}$
　　　　　　　　　　　eye-concave~RDP
　　　　　　　　　　　'concave eye; concentrate'
　　　　　　d.　　　　$hi^{21}$-$do^{33}$~$do^{33}$
　　　　　　　　　　　belly-out~RDP
　　　　　　　　　　　'pot-bellied'

*4.6. Totalitativity with Perfective Particle*

Reduplication with verbs can denote the meaning of totalitative, signalling that the action has been done completely and/or the referent has been totally affected. When reduplication is used to denote this meaning with verbs, the verb co-occurs with a series of postverbal particles $xu^{55}$ $tu^{21}$ $g\vartheta^{21}$ 'PFV PRF APPL'. Note that two aspect markers occur here: the perfective particle $xu^{55}$ and the perfect aspect particle $tu^{21}$, but the meaning of the co-occurrence of these two aspect particles in the series is the perfective one, i.e., a state resulting from a previous action that is still going on. The co-occurrence of these two aspect particles is not found anywhere else. The perfective meaning is indicated by the perfective particle, not the verbal reduplication, and the meaning will not change if the reduplication is not present. The occurrence of applicative particle $g\vartheta^{21}$ indicates that the referent is affected by the action.

| (39) | $\varepsilon u^{55}$ | $za^{13}$~$za^{21}$ | $xu^{55}$ | $tu^{21}$ | $g\vartheta^{21}$=$a^{21}$ |
|---|---|---|---|---|---|
| | wheat | RDP~cut | PFV | PRF | APPL=SFM |
| | 'the wheat has been all cut' | | | | |

| (40) | $\varepsilon u^{33}mi^{21}gu^{21}$ | $dzi^{33}$ | $ha^{33}dzi^{33}$ | $\varepsilon u^{13}$~$\varepsilon u^{21}$ | $xu^{55}$ | $tu^{21}$ | $g\vartheta^{21}$=$a^{21}$ |
|---|---|---|---|---|---|---|---|
| | pomegranate | TOP | mouse | RDP~eat | PFV | PRF | APPL=SFM |
| | 'pomegranate has been eaten up by a mouse' | | | | | | |

*4.7. Reduplication in Lexical Nominalisation*

Nominalisation in Kua'nsi can be categorised as two types, lexical and grammatical nominalisation. In lexical nominalisation, only one word is nominalised, while in grammatical nominalisation, a constituent or whole clause is nominalised. When the lexical nominalisation is derived by the manner nominaliser $su^{21}$, the nominalised word has to be reduplicated.

The nominal derived with the manner nominaliser $su^{21}$ indicates how a referent looks or how an action is carried out. In the lexical nominalisation with $su^{21}$, the nominaliser usually occurs after stative verbs and adjectives that denote colour, size, shape, or state, and it is obligatory for the verb or adjective to be reduplicated, as in (41). Reduplication in this lexical nominalisation does not encode intensification or the other functions of reduplication with adjectives and verbs, as mentioned above, but it is required by the nominalisation process.

(41)    a.    *na³³kwa³³~kwa³³*               *su²¹*

                    black~RDP               NMLZ.manner

                    'the black one'

          b.    *kʑ³³tʰʑ⁵⁵~tʰʑ⁵⁵*           *su²¹*

                    fat~RDP                 NMLZ.manner

                    'the fat one'

          c.    *ta⁵⁵la²¹zi³³~zi³³*          *su²¹*

                    lazy~RDP              NMLZ.manner

                    '(in) a lazy way'

          d.    *do³³tsʰi²¹~tsʰi²¹*         *su²¹*

                    inactive~RDP          NMLZ.manner

                    'inactively, in an inactive way'

The nominalised construction takes the place of nouns in the noun phrase or is used adverbially. For example, in (42), the lexical nominalised construction $si^{21}ni^{55}ni^{55}$ $su^{21}$ 'the red one' takes the place of the head noun in the noun phrase and functions as the core argument of the predicate. In (43), the nominal $bo^{33}$~$bo^{33}$ $su^{21}$ occurs before the predicate and is used adverbially to denote the manner of the action, i.e., how the speaker made the bed. Compared with the use of reduplication to derive adverbs, the manner nominalisation with $su^{21}$ is a more productive way in Kua'nsi to produce adverbial construction. Comparing (43) and (14), when the nominalised construction is larger than a single word, i.e., it is grammatical nominalisation, the reduplication is not involved in the process of nominalisation, but the nominalised constructions can be used adverbially in the same way.

(42)    *si²¹ɲi⁵⁵~ɲi⁵⁵*    *su²¹*        *su³³*    *tʑ²¹*    *i³³*      *tu²¹*

          red~RDP       NMLZ.manner    three    CLF:flower   bloom    PRF

          'three red flowers bloomed'

(43)    *bo³³~bo³³*    *su²¹*       *kʰa²¹*    *u³³*      *tɕja³³*

          flat~RDP      NMLZ.manner  cover     succeed    able.to

          'I can make (the bed layer) flat' (lit. 'I can spread the bed with sheet in a flat way')

## 5. Conclusions

This paper shows that, in Kua'nsi, a less morphologically rich language, reduplication is an important part of its grammar. This description of Kua'nsi reduplication contributes to and raises questions about how reduplication can be better understood cross-linguistically. Formally, Kua'nsi distinguishes three forms of reduplication, full, partial, and discontinuous. While some reduplicants are the exact copy of their base, others involve a tonal modification or addition of a new segment. In addition to the various forms of reduplication, another tier of the complexity of reduplication in Kua'nsi is its unpredictability of some forms of reduplication. As it is difficult to generalise a rule to predict the specific form of reduplication in some contexts, it will be interesting to see how speakers would reduplicate an unfamiliar form and how children learn these different patterns of reduplication.

Reduplication in Kua'nsi is employed to encode many functions. Some functions may be rare cross-linguistically because many Tibeto-Burman languages were not included in previous typological research. Some functions of Kua'nsi reduplication seem to be hard to account for with iconicity. As reduplication is common in many Sino-Tibetan languages, a better understanding of reduplication in the language family can contribute to the broad typology of reduplication.

More broadly, the case study of Kua'nsi reduplication has shown that Sino-Tibetan languages can contribute more to the broad typology of morphological diversity. A stereotype toward Sino-Tibetan languages is that they do not have or show a low degree of morphological complexity, and thus are not of interest to morphological research, owing to the strong influence of Mandarin Chinese. However, this is not true. There is a great degree

of morphological diversity in the language family from isolating languages like Mandarin Chinese to polysynthetic languages like Japhug. This diversity may not be apparent to people who are not familiar with Sino-Tibetan, especially Tibeto-Burman languages, and Mandarin Chinese is often selected as representative of Sino-Tibetan languages in typological research; for example, Lǐ and Ponsford (2018). This practice masks the intriguing diversity in the language family. Language samples should be more carefully selected for any typological research.

**Funding:** This research was funded by the ARC Centre of Excellence for the Dynamics of Language (Project ID: CE140100041).

**Institutional Review Board Statement:** This study was conducted in accordance with the Declaration of Helsinki, and approved by the Human Research Ethics Committee of the Australian National University (Protocol 2018/213) on 7th June 2018.

**Informed Consent Statement:** Informed consent was obtained from all subjects involved in this study.

**Data Availability Statement:** Data used in this article is archived with the Pacific and Regional Archive for Digital Sources in Endangered Cultures (PARADISEC). Available at http://catalog.paradisec.org.au/collections/HH1.

**Acknowledgments:** I would like to thank all the Kua'nsi people who generously and patiently shared their language with me. For this paper, I am particularly thankful to JIAO Xiongxin 绞雄心, who helped me collect additional data when travel was restricted during the COVID-19 pandemic. I would also like to thank the audience at ICSTLL54, Bethwyn Evans, Elvis Yang Huang and two reviewers for their valuable and constructive comments.

**Conflicts of Interest:** The author declares no conflict of interest. The funder had no role in the design of the study; in the collection, analysis, or interpretation of data; in the writing of the manuscript; or in the decision to publish the results.

## Abbreviations

| | |
|---|---|
| 1, 2, 3 | first, second, third person |
| ADDR | addressee |
| ASYM | asymmetric |
| AUG | augmentative |
| APPL | applicative |
| CLF | classifier |
| COMP | complementiser |
| COP | copula |
| DIST | distal |
| ECLM | exclamation |
| PRF | perfect aspect |
| PFV | perfective aspect |
| GEN | general |
| IMP | imperative |
| INFR | inferential evidential |
| NEG | negative |
| NMLZ | nominaliser |
| Q | question marker |
| PROX | proximal |
| RDP | reduplication |
| SBJV | subjunctive |
| SFM | sentence final marker |
| SG | singular |
| TOP | topic |
| TWD | towards |

## Notes

[1] In many Sino-Tibetan languages, the V-NEG-V construction is often used to denote polar questions. In some Ngwi languages, this construction and reduplication can be both used to indicate polar questions. However, the V-NEG-V is not attested in Kua'nsi. Also, there is no evidence that suggests a historical link between these two patterns in Kua'nsi.

[2] The numbers are used for the notation of tones in this paper. The number 1 represents a low tone, and the number 5 represents a high one. Two digits in sequence indicate starting and ending pitches.

[3] As a reviewer suggested, the rare tonal pattern can be explained by the functional load. This tone /13/ is rare, so it is only available to occur in limited constructions without causing widespread homophony and ambiguity. This can is a potential explanation for why /13/ tone is associated with the augmentative particle and reduplication, both of which can be used to emphasise or intensify the degree.

[4] The glottal stop /ʔ/ can occur as syllable onset in Kua'nsi and form syllables like the sequence /ʔV/. However, it is possible to drop the syllable onset /ʔ/ when the syllable is word-initial, and there is no contrast between the syllable with or without the glottal stop. For example, both $ʔu^{33}nu^{21}$ and $u^{33}nu^{21}$ mean 'dog'.

[5] As a review suggested, the syllabic nasal does not participate in reduplication can also be explained by the concept of functional load. As the syllabic nasal on verbs and adjectives denotes the meaning of negation, its reduplication would express the opposite meaning and so its function prevents it from being reduplication. This can be a possible explanation for this, but I do not have more convincing evidence to support this idea.

[6] All plural pronouns in Kua'nsi have /21/ tone and their possessive forms are the same. For example, $ŋu^{21}$ '1PL' or '1PL.POSS'; $ɲi^{21}$ '2PL' or '2PL.POSS'; $i^{21}$ '3PL' or '3PL.POSS'. An alternative analysis is that the plural pronouns also undergo the same tonal inflection as singular pronouns. However, as their tones are already /21/, the surface forms do not change.

[7] The case marking clitic $=a^{21}$ is the only case marker in Kua'nsi and it can mark a range of arguments. It can mark the S and A argument when one of the verbs in the predicate denotes the change of internal state of the S or A argument, such as $ts^hu^{55}$ '(feel) hot', $hi^{21}mɤ^{33}$ 'hungry' and $xwa^{21}$ 'thirsty'. It also marks the P or more patient-like argument of the transitive predicate. So sometimes, both A and P arguments are marked by the case marking clitic. If there are three core arguments as ditransitive predicates, it is the recipient or goal argument is marked. It is also used to mark the locative oblique. The case marking is sometimes optional depending on semantic and pragmatic factors, such as animacy and topicality.

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
