# Peer review of "Reduplication in Kua’nsi"

_languages, doi:10.3390/languages8020130_

Round 1

Reviewer 1 Report

See attached file.

Reviewer 2 Report

The main flaw in this paper is that the Author does not clearly state the source of her/his language data. If they come from fieldwork, then a summary description of the database and of the speakers is in order.

Some specific comments:

On p. 1, the Author writes that “As the 30 description of reduplication implies, it is considered a morphological process in this paper, and so, by definition, any syntactic repetitions of words or phrases are excluded from consideration.” However, a few lines below s/he mentions full reduplication as a subtype of reduplication. It is thus unclear how s/he distinguishes full reduplication of words in morphology and in syntax. The first two examples in (1) could be argued to be syntactic, for instance, depending on the criteria applied.

P. 2: “For example, some Ngwi languages, including Kua’nsi, use verbal reduplication to form polar questions” – could it be the case that this pattern emerged from the loss of the negator in a verb – neg – verb construction? The A – non – A pattern is widespread in Sinitic, for instance.

P. 3: “One feature of reduplication in Kua’nsi is that although it is productive in the language, it is also lexicalised.” – I guess what the author means here is that there it is  productive, but the patterns are apparently arbitrary?

P. 3: Please explain the number-based tone notation (not obvious to non-sinotibetanists).

P. 3: (3) are not all minimal pairs.

P. 3: Thus, the structure of Kua’nsi words is often to be > said to be

P. 3 : Based on (4), it seems that the Author treats [j] as a consonant: why?

P. 4: The beginning of Section 3 overlaps with the Introduction.

P. 5: “A single consonant or vowel cannot be the base of reduplication.” – thus, not all syllables can be reduplicated.

Table 1: the base (non-reduplicated ) forms are necessary to understand the patterns.

P. 7: “It might be possible that monosyllabic words rather  show the same reduplication process as initial syllable reduplication, as the tone change can occur with some monosyllabic words.” – I must admit I did not understand what the Author means here.

P. 10: Is this pattern of incorporation with body parts productice in the language?

(29) – (30): Why is the perfect aspect particle glossed as ‘put’?

P. 11: “A stereotype toward Sino-Tibetan languages is that they do not have any morphology and thus are not of interest to morphological research. However, this is not true. There is a great degree of morphological diversity in the language family from isolating languages like Mandarin Chinese to polysynthetic languages like Japhug. Mandarin Chinese is often se lected as representative of Sino-Tibetan languages in typological research, but this practice masks the intriguing diversity in the family.” – I think the Author here confuses ‘Sinotibetan’ with ‘Sinitic’. It is Sinitic, not Sinotibetan as a whole, which is considered as having no morphology; and Mandarin is often (mis)taken as representative of Sinitic, not of Sinotibetan.

Round 2

Reviewer 1 Report

the revisions have sparked additional issues that need to be addressed.

REVIEWER COMMENTS 2197178 v2
Overview
I thank the author for considering the previous comments, and incorporating those that were deemed relevant. I believe the revised manuscript has been substantively improved. These revisions have surfaced a few additional comments, though; see below.

Comments
p. 4, ln 200-02 (point “c”): It is explained that tonal inflection only occurs with one morpheme, the 1SG PRO. If this is the only instance in the language where this occurs, then tonal inflection is not productive and it would be better not to mention it. The similarity of the 1SG & 1SG.POSS morphemes may be due to historical processes or even coincidence; it doesn’t seem relevant to reduplication. Or, if the author feels strongly it should be mentioned, a footnote would be more appropriate.

p. 4, ln 203 onwards (Sec. 2.4): The discussion of word classes is inconsistent and a bit confusing. I have four related comments here:

(1) On p. 5 (ln 211-2) it is stated that “Kua’nsi fundamentally distinguishes three major word classes, nouns, verbs and adjectives.” This suggests that there are only three classes and no more. However, grammatical particles likely form a separate class, and certain adverbs --such as intensifiers as well as sentential adverbs like ‘also’ – probably also need their own category. These need to be tested and clarified in the article.

(2) So-called “adjectives” in Ngwi languages are typically stative verbs rather belongint to a separate word class. The discussion (p. 5, ln 257) shows that they indeed share every feature of verbs, with the additional ability to directly modify nouns. This suggests that “adjectives/stative verbs” are in fact a sub-category of verbs rather than a completely different category.

(3) Later, adverbs are described as a word class (p. 6 ln 259), though they were not included in the initial list on p. 5. Cross-linguistically, the adverb category is usually a mix of disparate items, including nouns that function as temporal/locative adverbs, unambiguous adverbs such as intensifiers, plus constructions that consist of or derive from “adjectives”. This needs more explanation in Sec. 2.4 so that the functions described in Sec. 4.4 are clearer.

(4) Sec. 4.4 asserts that reduplicated stative verbs change class from “adjective” to “adverb”, but the evidence presented does not support that idea. When they reduplicate to communicate intensification (Sec. 4.2), the reduplicative process does not trigger a change in category. In ex. 35 & 36, we likewise see reduplication without any other change, which suggests that no word class conversion occurs here either. There are a number of languages around the world in which ADJ perform an adverbial function without changing word class – the interpretation is pragmatic rather than syntactic – and this analysis may be relevant here.

p. 5, ln 210 (and elsewhere): The case marking clitic =a21 is mentioned, but not its specific function. Not all Ngwi languages have case markers, so it would be useful to know which cases the particle marks.

p. 9 (the PDF erroneously says p. 2), ln 448-458: To add to this prosodic idea, it may be that the goal of the prosodic construction is a combination of a HIGHER tone (e.g. high or rising) followed by a LOWER tone (i.e. lower than the previous one). If this is the case, syllables with tone 55 don’t change because they already fit the pattern and don’t need further modification.
This analysis reduces the irregularities in the pattern, but doesn’t explain all of them, i.e. why 33-33 combinations don’t change. I agree that many of these have become routinized, if not completely lexicalized, which likely explains some of the messiness.

p. 10 (the PDF erroneously says p. 3), ln 518, ex. 27 (and elsewhere): AUG would be a more intuitive way to abbreviate ‘augmentative’. AGMT could be mistakenly interpreted as ‘argument’ or ‘agreement’. Also, in this example the asterisk before lwa13 seems to indicate that its presence is not possible, but that contradicts what you say in the preceding paragraph (ln 523-5).

p. 12 (the PDF erroneously says p. 5), ln 635-46: The discussion here is not clear. It says that the reduplicated constructions in ex. 40a & b are used to “modify nouns”, but they appear to be nominals that can take the place of nouns, i.e. serving as core arguments without additional elements. Is that the case? Examples of these constructions in clauses would be helpful here.

p. 12 (the PDF erroneously says p. 5), ln 645-46: The final sentence here is vague; it would be better to state outright what function you mean to exclude here.

Also, the revisions have introduced a number of English language errors. I assume the journal’s editor will work with the author to address them.
